# Differentiable Simulation of Soft Multi-body Systems

**Yi-Ling Qiao** [*]
University of Maryland, College Park

**Junbang Liang** [*]
University of Maryland, College Park

**Vladlen Koltun**
Intel Labs

**Ming C. Lin**
University of Maryland, College Park

## Abstract

We present a method for differentiable simulation of soft articulated bodies. Our work enables the integration of differentiable physical dynamics into gradient-based pipelines. We develop a top-down matrix assembly algorithm within Projective Dynamics and derive a generalized dry friction model for soft continuum using a new matrix splitting strategy. We derive a differentiable control framework for soft articulated bodies driven by muscles, joint torques, or pneumatic tubes. The experiments demonstrate that our designs make soft body simulation more stable and realistic compared to other frameworks. Our method accelerates the solution of system identification problems by more than an order of magnitude, and enables efficient gradient-based learning of motion control with soft robots.

## 1 Introduction

Soft articulated bodies have been studied and utilized in a number of important applications, such as microsurgery [32], underwater robots [37], and adaptive soft grippers [20]. Since the compliance of deformable materials can enable robots to operate more robustly and adaptively, soft biomimetic robots are drawing a lot of attention and have made considerable progress. A snailfish robot dives at a depth of 10,900 meters in the Mariana Trench [43]. Drones equipped with soft manipulators grasp and transmit objects with a 91.7% success rate [17]. Soft hands with pneumatic actuators are able to grasp objects of different shapes, including water bottles, eyeglasses, and sheets of cloth [12]. To enable rapid prototyping of soft robots and efficient design of control algorithms through virtual experiments, we aim to create a realistic deformable multi-body dynamics framework, in which soft articulated robots can be simulated to learn powerful control policies.

Design and control of soft robots are challenging because of their nonlinear dynamics and many degrees of freedom. Differentiable physics has shown great promise to deal with such complex problems [4, 14, 31, 68]. One possibility is to treat soft bodies as volumes that are modeled as sets of particles or finite elements [29, 16]. These methods have made great progress, but the volumetric representations are difficult to scale to large multi-body systems and are poorly suited to modeling internal skeletons. Moreover, contact handling in recent differentiable physics frameworks [51, 30] often does not comply with Coulomb's Law, which is central to plausible visual realism and correct physical behavior.

In this paper, we design a powerful and accurate differentiable simulator for soft multi-body dynamics. Since our entire framework is differentiable, our method can be embedded with gradient-based optimization and learning algorithms, supporting gradient-based system identification, motion planning, and motor control. Within the simulator, we first use tetrahedral meshes to enable *adaptive resolution* and more accurate modeling. Next, to couple soft materials with articulated skeletons, we design a

---

[*]Equal contribution.

35th Conference on Neural Information Processing Systems (NeurIPS 2021).

*top-down matrix assembly algorithm* within the local steps of Projective Dynamics [5]. For accurate contact handling, we extend and generalize a dry friction model previously developed for cloth simulation [54] to soft solids and introduce a new *matrix splitting strategy* to stabilize the solver. In addition, our simulation framework incorporates actuator models widely used in robotics, including muscles [40], joint torques [75], and pneumatic actuators [12]. With the support of the articulated skeleton constraints, dry frictional contact, and versatile actuators, our novel differentiable algorithm can simulate soft articulated robots and compute gradients for a wide range of applications.

The key contributions of this work are as follows.

- A top-down matrix assembly algorithm within Projective Dynamics to make soft-body dynamics compatible with reduced-coordinate articulated systems (Sec. 4).
- An extended and generalized dry friction model for soft solids with a new matrix splitting strategy to stabilize the solver (Sec. 5).
- Analytical models of muscles, joint torques, and pneumatic actuators to enable more realistic and stable simulation results (Sec. 4.3 and Appendix C & D).
- A unified differentiable framework that incorporates skeletons, contact, and actuators to enable gradient computation for learning and optimization (Sec. 6).
- Experimental validation demonstrating that differentiable physics accelerates system identification and motion control with soft articulated bodies up to *orders of magnitude* (Sec. 6).

In the following paper, Section 2 will discuss related papers in deformable body simulation and differentiable physics. Section 3 is basically a preliminary of projective dynamics to explain the high-level simulation framework and define notations that will be used later. Section 4 and 5 are about how to deal with articulated skeleton and contact in our method. Section 6 will show the ablation studies and comparisons results with other learning methods. Code is available on our project page: `https://github.com/YilingQiao/diff_fem`

## 2 Related Work

**Deformable body simulation** using Finite Element Method (FEM) plays an important role in many scientific and engineering problems [32, 20, 57]. Previous works model soft bodies using different representations and methods for specific tasks. There are several kinds of approaches for modeling body actuation. Pneumatic-based methods [9] change the rest shape to produce reaction forces. Rigid bones attached within soft materials are also used to control the motion of deformable bodies [52, 38, 18, 45]. To further simulate biologically realistic motion, it is common to apply joint torques in articulated skeletons [36]. For example, [33, 76] use articulated body dynamics to govern the motion while handling collisions using soft contact. Inspired by animals, different designs of muscle-like actuators for soft-body simulations were also proposed [41, 1, 42].

Regarding contact modeling, spring-based penalty forces are widely used [58, 67, 27] for their simplicity. More advanced algorithms include inelastic projection [6, 24] and barrier-based repulsion [47, 48]. However, these methods do not always conform to Coulomb's frictional law. We opt for a more realistic dry frictional model [44] to better handle collisions.

Projective Dynamics [5] is widely used for its robustness and efficiency for implicit time integration. It has been extended to model muscles [59], rigid skeletons [46], realistic materials [53], and accurate contact forces [54]. Our method also adopts this framework for faster and more stable time integration. In contrast to the aforementioned methods using Projective Dynamics, our algorithm is the first to enable joint actuation in articulated skeletons together with a generalized dry frictional contact for soft body dynamics.

**Differentiable physics** has recently been successfully applied to solve control and optimization problems. There are several types of physically-based simulations that are differentiable, including rigid bodies [10, 11], soft bodies [30, 29, 39, 19, 16], cloth [51, 62], articulated bodies [21, 74, 63], and fluids [71, 73, 28, 70]. Differentiable physics simulation can be used for system identification [68, 26], control [69], and design [14, 49]. For differentiable soft-body dynamics, Du et al. [16] propose a system for FEM simulation represented by volume mesh. This system has been applied to robot design [56] and control [15]. Different from this work [16], our approach uses tetrahedral meshes with adaptive resolution to model finer detail and scale better to complex articulated bodies.

Hu et al. [30], Krishna Murthy et al. [39] use source code transformation to differentiate the dynamics, but their contact model does not follow Coulomb's law. Geilinger et al. [19] simulate soft materials attached to rigid parts with penalty-based contact force, but their use of maximal coordinates makes it difficult to incorporate joint torques. In comparison, our model has realistic contact handling, versatile actuators, and skeletons with joint constraints, thereby enabling our method to simulate a much wider range of soft, multi-body systems not possible before.

There are other works that approximate physical dynamics using neural networks [50, 2, 72, 65]. These methods are inherently differentiable but cannot guarantee physical correctness outside the training distribution.

## 3    Soft Body Simulation Using Projective Dynamics

We use Projective Dynamics [5] to model the physics of soft, multi-body systems because its efficient implicit time integration can make the simulation more stable. We briefly introduce Projective Dynamics below. The dynamics model can be written as

$$\mathbf{M}(\mathbf{q}_{n+1} - \mathbf{q}_n - h\mathbf{v}_n) = h^2(\nabla E(\mathbf{q}_{n+1}) + \mathbf{f}_{ext}), \tag{1}$$

with $\mathbf{M}$ being the mass matrix, $\mathbf{q}_n$ the vertex locations at frame $n$, $h$ the time step, $\mathbf{v}_n$ the velocity, $E$ the potential energy due to deformation, and $\mathbf{f}_{ext}$ the external forces. We choose implict Euler for a stable time integration, then the state $\mathbf{q}_{n+1}$ can be solved by

$$\mathbf{q}_{n+1} = \arg\min_{\mathbf{q}} \frac{1}{2h^2}(\mathbf{q} - \mathbf{s}_n)^\top \mathbf{M}(\mathbf{q} - \mathbf{s}_n) + E(\mathbf{q}), \tag{2}$$

where $\mathbf{s}_n = \mathbf{q}_n + h\mathbf{v}_n + h^2\mathbf{M}^{-1}\mathbf{f}_{ext}$. Projective Dynamics reduces the computational cost by introducing an auxiliary variable $\mathbf{p}$ to represent the internal energy as the Euclidean distance between $\mathbf{p}$ and $\mathbf{q}$ after a projection $\mathbf{G}$:

$$E(\mathbf{q}) = \sum_i \frac{\omega_i}{2} \|\mathbf{G}_i\mathbf{q} - \mathbf{p}_i\|_F^2, \tag{3}$$

where $\omega$ is a scalar weight, and $E$ contains internal energy from different sources, such as deformations, actuators, and constraints. The computation of $\omega$, $\mathbf{G}$, and $\mathbf{p}$ is dependent on the form of the energy. Combining all energy components into Eq. 2, we have

$$\mathbf{q}_{n+1} = \arg\min_{\mathbf{q}} \frac{1}{2}\mathbf{q}^\top \left(\frac{\mathbf{M}}{h^2} + \mathbf{L}\right)\mathbf{q} + \mathbf{q}^\top \left(\frac{\mathbf{M}}{h^2}\mathbf{s}_n + \mathbf{Jp}\right), \tag{4}$$

where $\mathbf{L} = \sum \omega_i \mathbf{G}_i^\top \mathbf{G}_i$ and $\mathbf{J} = \sum \omega_i \mathbf{G}_i^\top \mathbf{S}_i$, and $\mathbf{S}_i$ is the selector matrix. Since this is a quadratic optimization without constraints, its optimal point is given by the solution of the following linear system:

$$\left(\frac{\mathbf{M}}{h^2} + \mathbf{L}\right)\mathbf{q}_{n+1} = \frac{\mathbf{M}}{h^2}\mathbf{s}_n + \mathbf{Jp} \tag{5}$$

Note that the estimation of $\mathbf{p}$ is based on the current values of $\mathbf{q}$. Therefore we need to alternate between computing $\mathbf{p}$ and solving $\mathbf{q}$ until convergence. Luckily, both steps are fast and easy to solve: solving $\mathbf{q}$ in Eq. 5 is easy because it is a simple linear system, and computing $\mathbf{p}$ can be fast because it is local and can be parallelized. We show the generic Projective Dynamics method in Alg. 1.

---

**Algorithm 1** Soft body simulation using Projective Dynamics

---
1: $\mathbf{x}^1 \leftarrow$ initial condition
2: **for** $t = 1$ **to** $n - 1$ **do**
3:     **while** not converged **do**
4:         Compute $\mathbf{p}_i$ for all energy components $i$ according to $\mathbf{q}$ (Local step)
5:         Solve $\mathbf{q}$ in Eq. 5 according to $\mathbf{p}_i$ (Global step)
6:     **end while**
7: **end for**

---

# 4 Articulated Skeletons

Skeletons are indispensable for vertebrate animals and nowadays articulated robots. However, adding skeletons into the soft body simulation is challenging. First, the rigid bones cannot be simply replaced by soft materials with large stiffness, since this can make the system unstable and unrealistic. Second, joint connections between bones must be physically valid at all times, and thus also cannot be modeled as soft constraints. Moreover, the formulation should support joint actuation as torques to drive the multi-body system like an articulated robot. Li et al. [45] proposed a method for passive articulated soft-body simulation with ball joint constraints. We extend this method to enable rotational/prismatic joints, torque actuation, and precise joint connections without introducing extra constraints.

## 4.1 Rigid Body System

When integrated with hard skeletons, vertices on the rigid parts can be expressed as

$$\mathbf{q}_k = \mathbf{Q}\mathbf{T}_k^r\mathbf{V}_k, \tag{6}$$

where $\mathbf{Q} = (\mathbf{I} \quad \mathbf{0})$ is the projection from homogeneous coordinates to 3D coordinates, $\mathbf{T}_k^r \in \mathbb{R}^{4\times4}$ the rigid transformation matrix, and $\mathbf{V}_k \in \mathbb{R}^{4\times m_k}$ the rest-pose homogeneous coordinates of the $k^{\text{th}}$ rigid body.

During the global step in Projective Dynamics, we do not directly solve for $\mathbf{T}_k^r$, but for the **increment** $\Delta\mathbf{z}_k$ in its degree-of-freedom (DoF), to avoid nonlinearity. This formulation restricts the changes of the rigid vertices to the tangent space yielded by the current $\mathbf{T}_k^r$:

$$\mathbf{q}_k^{i+1} = \mathbf{q}_k^i + \Delta\mathbf{q}_k^i \approx \mathbf{q}_k^i + \frac{\partial\mathbf{q}_k^i}{\partial\mathbf{z}_k}\Delta\mathbf{z}_k^i, \tag{7}$$

where $\mathbf{q}_k^i$ are the vertex locations of the $k^{\text{th}}$ body in the $i^{\text{th}}$ iteration step, and $\mathbf{z}_k$ is the variable defining the DoF of the $k^{\text{th}}$ rigid body, including the rotation and the translation. The nonrigid part of the vertices can also be integrated with this formulation simply with $\frac{\partial\mathbf{q}^i}{\partial\mathbf{z}} = \mathbf{I}$.

Let $\mathbf{B} = \frac{\partial\mathbf{q}^i}{\partial\mathbf{z}}$ be the Jacobian of the concatenated variables. Eq. 4 can be rewritten as

$$\Delta\mathbf{z}^i = \arg\min_{\Delta\mathbf{z}} \frac{1}{2}\Delta\mathbf{z}^\top\mathbf{B}^\top\left(\frac{\mathbf{M}}{h^2} + \mathbf{L}\right)\mathbf{B}\Delta\mathbf{z} + \Delta\mathbf{z}^\top\mathbf{B}^\top\left(\left(\frac{\mathbf{M}}{h^2} + \mathbf{L}\right)\mathbf{q}^i - \left(\frac{\mathbf{M}}{h^2}\mathbf{s}_n + \mathbf{J}\mathbf{p}\right)\right) \tag{8}$$

After solving $\Delta\mathbf{z}^i$, the new vertex states $\mathbf{q}^{i+1}$ are derived by the new rigid transformation matrix $\mathbf{T}_k^{r\prime}$ using Eq. 6, which is subsequently computed in the local step, discussed below.

**Local step.** The variable $\Delta\mathbf{z}_k^i$ is composed of the increment of rotation $\omega_k^i$ and translation $l_k^i$ based on the current transformation $\mathbf{T}_k^r$:

$$\Delta\mathbf{q}_k^i = \begin{bmatrix}-[\mathbf{q}_k^i]_\times & \mathbf{I}\end{bmatrix}\begin{bmatrix}\omega_k^i \\ l_k^i\end{bmatrix}. \tag{9}$$

Here $[\mathbf{q}_k^i]_\times$ is defined as the vertical stack of the cross product matrices of all vertices in the $k^{\text{th}}$ rigid body. During the local step, we compute the SVD of the new transformation matrix after integrating $\omega_k^i$ and $l_k^i$,

$$\mathbf{T}_k = \begin{bmatrix}\mathbf{I} + \omega_k^{i*} & \mathbf{0} \\ \mathbf{0} & 1\end{bmatrix}\mathbf{T}_k^r + \begin{bmatrix}\mathbf{0} & l_k^i \\ \mathbf{0} & 0\end{bmatrix} = \mathbf{U}\Sigma\mathbf{V}^\top, \tag{10}$$

and restrict it to SO(3) to obtain the new rigid transformation, $\mathbf{T}_k^{r\prime} = \mathbf{U}\mathbf{V}^\top$.

The local step of Projective Dynamics for a single rigid body is the same as in [46]. However, we propose Eq. 7 to generalize the coupling to kinematic trees [55] with precise and actuated articulation.

## 4.2 Top-down Matrix Assembly for Articulated Body Systems

The articulated body system formulation is similar to the rigid body one, except that the transformation matrix is now chained:

$$\mathbf{T}_k^r = \prod_{u\in U_k}\mathbf{A}_u, \tag{11}$$

where $U_k$ contains all ancestor of the $k^{\text{th}}$ link (inclusive), and $\mathbf{A}_u$ is the local transformation matrix defined by joint $u$.

For rigid bodies, the vertex locations of a rigid body only depend on the body's own DoF variables. In the articulated system, however, they are also affected by the body's ancestors. Therefore, $\mathbf{B}$ changes from a block diagonal matrix to a block lower triangular matrix if the rigid body vertices are ordered by their kinematic tree depth.

To compute the matrix $\mathbf{B}$, we consider a link $u$ with one of its non-root ancestor $v$. By the definitions in Sec. 4.1, the corresponding block in matrix $\mathbf{B}$ is

$$\mathbf{B}_{u,v} = \frac{\partial \mathbf{T}_u^r \mathbf{V}_u}{\partial \mathbf{z}_v} = \mathbf{Q}\mathbf{P}_v \frac{\partial \mathbf{A}_v}{\partial \mathbf{z}_v} \mathbf{S}_{v,u} \mathbf{V}_u, \tag{12}$$

where $\mathbf{P}_v$ is the prefix product of the local transformation matrices of the link chain from root to $v$ (exclusive), and $\mathbf{S}_{v,u}$ is the suffix product from $v$ to $u$. In boundary cases where $u = v$, the formulation becomes

$$\mathbf{B}_{u,u} = \mathbf{Q}\mathbf{P}_u \frac{\partial \mathbf{A}_u}{\partial \mathbf{z}_u} \mathbf{V}_u. \tag{13}$$

When $v$ is the root and thus has the same DoFs as a rigid body, using the results from Sec. 4.1, the formulation can be simplified to

$$\mathbf{B}_{u,root} = \mathbf{Q}\left[-[\mathbf{q}_u]_\times \quad \mathbf{I}\right]. \tag{14}$$

Computing Eq. 12 requires the matrix products $\mathbf{P}_v$ and $\mathbf{S}_{v,u}$ of a link chain $(v, u)$ in the tree. Straightforward approaches here could result in $O(N^3)$ complexity, where $N$ is the number of links. However, by utilizing the kinematic tree and conducting the computation in top-down order, the complexity can be reduced to $O(N^2)$, which is optimal. The key observation is that the prefix and suffix products can be computed recursively:

$$\mathbf{P}_v = \mathbf{P}_{v'}\mathbf{A}_{v'} \tag{15}$$
$$\mathbf{S}_{v',u} = \mathbf{A}_v\mathbf{S}_{v,u}, \tag{16}$$

assuming $v'$ is the parent link of $v$. When we traverse the kinematic tree in a depth-first order, the prefix product can be computed in $O(1)$. The suffix product is also obtained as we iterate along the path back to the root. Algorithm 2 shows the matrix assembly method starting from the root node:

---
**Algorithm 2** Matrix Assembly for the Articulated System

---
1: Input: tree link $u$
2: Compute $\mathbf{P}_u$ using Eq. 15
3: $v \leftarrow u$
4: **while** $v$ is not root **do**
5:     Compute $\mathbf{S}_{v,u}$ using Eq. 16
6:     Compute $\mathbf{B}_{u,v}$ using Eq. 12
7:     $v \leftarrow \text{parent}(v)$
8: **end while**
9: Compute $\mathbf{B}_{u,root}$ using Eq. 14
10: **for** $s$ **in** descendants$(u)$ **do**
11:     Solve link $s$ recursively
12: **end for**

---

The transformation matrix $\mathbf{A}$ and the Jacobian of a joint depend on the joint type. This is further derived in Appendix C.

## 4.3 Articulated Joint Actuation

Eq. 8 is a quadratic optimization, so the optimal $\Delta \mathbf{z}^i$ is given by the linear system

$$\mathbf{H}\Delta \mathbf{z}^i = \mathbf{k}, \tag{17}$$

where $\mathbf{H} = \mathbf{B}^\top \left( \frac{\mathbf{M}}{h^2} + \mathbf{L} \right) \mathbf{B}$ and $\mathbf{k} = -\mathbf{B}^\top \left( \left( \frac{\mathbf{M}}{h^2} + \mathbf{L} \right) \mathbf{q}^i - \left( \frac{\mathbf{M}}{h^2} \mathbf{s}_n + \mathbf{J}\mathbf{p} \right) \right)$. Reordering the vertices into sets of deformable and rigid ones yields the following partitioning of the matrix:

$$\begin{bmatrix} \mathbf{H}_d & \mathbf{H}_c^\top \\ \mathbf{H}_c & \mathbf{H}_r \end{bmatrix} \begin{bmatrix} \Delta \mathbf{z}_d^i \\ \Delta \mathbf{z}_r^i \end{bmatrix} = \begin{bmatrix} \mathbf{k}_d \\ \mathbf{k}_r \end{bmatrix}, \tag{18}$$

where $*_d$ and $*_r$ represents the deformable and the rigid parts, respectively. The joint actuation can be directly added to $\mathbf{k}_r$ since the linear system is analogous to the basic formulation $\mathbf{M}\mathbf{a} = \mathbf{f}$ where the right hand side represents the sum of forces and/or torques. The formulation of pneumatic and muscle actuators can be found in Appendix D.

## 5 Contact Modeling

We handle the contact using Coulomb's frictional law via a Jacobian. To compute the velocities after collisions, we split the left-hand side of Equation 5 into the diagonal mass matrix $\mathbf{M}$ and the constraint matrix $h^2 \mathbf{L}$, and move the latter to the right-hand side:

$$\mathbf{M}\mathbf{v}^{i+1} = \mathbf{f} - h^2 \mathbf{L}\mathbf{v}^i + \xi^i, \tag{19}$$

where $\mathbf{f} = \mathbf{M}\mathbf{s}_n - (\mathbf{M} + h^2 \mathbf{L})\mathbf{q}_n + h^2 \mathbf{J}\mathbf{p}$ and the contact force $\xi^i$ is determined according to $\mathbf{f} - h^2 \mathbf{L}\mathbf{v}^i$ (the current momentum) to enforce non-penetration and static/sliding friction. The idea here is to enforce Coulomb's law at every iteration, which is ensured by solving $\mathbf{v}^{i+1}$ using the inverse of a diagonal matrix $\mathbf{M}$. As long as the solver converges at the end, the final $\mathbf{v}$ and $\xi$ will conform to the frictional law.

This method works well for cloth contacts [54], but cannot be directly applied to soft bodies, because solid continuum is much stiffer than thin sheets, i.e. the elements in $h^2 \mathbf{L}$ on the right-hand side are much larger than those in $\mathbf{M}$ on the left-hand side, resulting in severe oscillation or even divergence during the iterative solve.

We show that in order to guarantee the convergence of Equation 19, the time step $h$ has to satisfy a certain condition:

**Proposition 1.** *Assuming* $\mathbf{f}$ *and* $\xi$ *are fixed, Equation 19 converges if the time step* $h$ *satisfies*

$$h^2 < \frac{\rho}{24\sqrt{3}T\mu \sum_{k=1}^{3} \|\mathbf{q}_k - \mathbf{q}_0\|_2^2} \tag{20}$$

*where* $\rho$ *is the density,* $\mu$ *is the stiffness,* $T$ *is the number of tetrahedra, and* $\mathbf{q}_i$ *are the vertex positions.*

Details of the proof can be found in Appendix A. Using the setting in our experiments, where $T \approx 1000$, $\mu \approx 3 \times 10^5$, $\|\mathbf{q}_k - \mathbf{q}_0\|_2 \approx 10^{-2}$, and $\rho \approx 1$, we would need to set $h < 1/1934$ in order to ensure the convergence, which is too strict for the simulation to be useful in general applications.

**Splitting scheme.** We improve this method to be compatible with soft body dynamics by introducing a new splitting scheme. Eq. 19 is reformulated as

$$(\mathbf{M} + h^2 \mathbf{D})\mathbf{v}^{i+1} = \mathbf{f} - h^2 (\mathbf{L} - \mathbf{D})\mathbf{v}^i + \xi^i, \tag{21}$$

where $\mathbf{D}$ are the diagonals of $\mathbf{L}$. Our key observation is that (a) the diagonals of $\mathbf{L}$ are necessary and sufficient to stabilize the Jacobian iteration, and (b) adding extra diagonal elements to the left-hand side will not break the Coulomb friction law. We show in Appendix B that under the same assumption as Proposition 1, our method is guaranteed to converge no matter how big $h$ is. This improvement accelerates the simulation since larger time steps mean faster computation.

We also note that the new splitting scheme will not modify the behavior of the collision response because the convergence point of Eq. 19 is the same as that of Eq. 21, and thus Coulomb's Law is still satisfied at convergence.

## 6 Experiments

In this section, we first introduce our implementation and then report ablation studies that demonstrate the importance of skeletons and collision contacts in soft-body dynamics. Subsequently, we use the

gradients computed by our method to perform system identification; specifically, we estimate the physical parameters of bridges. Finally, we perform gradient-based learning of grasping and motion planning on robots with various actuators, including a pneumatic gripper, an octopus with muscles, and a skeleton-driven fish. Our method can converge more than an order of magnitude faster than reinforcement learning and derivative-free baselines.

## 6.1 Implementation

Our simulator is written in C++, the learning algorithms are implemented in PyTorch [60], and Pybind [34] is used as the interface. We run our experiments on two desktops, one with an Intel Xeon W-2123 CPU @ 3.6GHz and the other with an Intel i9-10980XE @ 3.0GHz, respectively. For differentiation, the numerical data structure in our simulator is templatized and integrated into the C++ Eigen library, such that our method can conveniently interoperate with autodiff tools to differentiate the dynamics. Our method can also run in pure C++ to perform forward simulation. We refer to the open-source code from [59] (Apache-2.0), [54] (GNU GPL v3.0), and [45] (MPL2) during our implementation. More details can be found in our code in the supplement.

To further improve the memory efficiency, we introduce a check-pointing scheme [7] into our pipeline. Instead of storing the entire simulation history, we only store the system's state in each step. During the backward pass, we reload the saved state vector and resume all the intermediate variables before the backpropagation. This strategy can save a major part of the memory, compared to the brute-force implementation. We conduct an experiment to compare the memory consumption with and without this check-

Table 1: Memory footprint (GB).

| steps | w/o ckpt | w/ ckpt |
|---|---|---|
| 10 | 0.9 | 0.1 |
| 20 | 1.4 | 0.1 |
| 100 | 6.9 | 0.1 |
| 200 | 15.7 | 0.1 |

pointing scheme. The results are reported in Table 1. CppAD [3] is used to differentiate the simulation here. In this experiment, we simulate a bridge and estimate its material properties as shown in Figure 3(a). The results show that the memory footprint of the baseline scales linearly with simulation length, while our checkpointing scheme keeps memory consumption nearly constant.

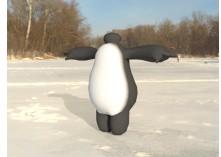

| | Bone length error | Body deformation error | Joint angle error | Runtime (ms) |
|---|---|---|---|---|
| No skeleton | $0.13 \pm 0.02$ | $0.20 \pm 0.14$ | $2.45 \pm 0.87$ | $132 \pm 2$ |
| Rigid [62] | **0** | $0.07 \pm 0$ | $1.81 \pm 0$ | $727 \pm 258$ |
| Passive [45] | **0** | $\mathbf{0.04 \pm 0.03}$ | $2.34 \pm 0.78$ | $163 \pm 4$ |
| MPM [30] | $0.48 \pm 0.32$ | $0.12 \pm 0.03$ | NaN | $\mathbf{11 \pm 0}$ |
| Ours | **0** | $\mathbf{0.05 \pm 0.03}$ | $\mathbf{0.41 \pm 0.30}$ | $191 \pm 4$ |

Figure 1: Ablation study of skeleton realization. 'Bone length error' measures the length change of a rigid bone, 'body deformation error' measures the deviation from the desired body length, and 'joint angle error' is the deviation from the target joint configuration. For all metrics, lower is better and 0 in error indicates the highest accuracy possible. Our model is the most physically realistic.

## 6.2 Ablation Study

**Skeleton constraints.** Controlling soft characters via skeletons is natural and convenient: vertebrate animals are soft, but are driven by piecewise-rigid skeletons. Our simulator supports skeletons and joint torques within soft bodies. This ablation study compares other designs with ours. In this experiment, a Baymax model [13] in its T-pose is released from above the ground, as shown in Figure 1. We embed 5 bones inside Baymax (4 in arms and legs, and 1 in the torso). When Baymax falls to the ground, we also add torques on its shoulders so it can lift its arms to a target Y-pose. More details of the setting and qualitative results can be found in Appendix E and the supplementary video. Three metrics, summarized in Figure 1, are used to measure realism. The metrics are averaged over 5 repetitions with different initial positions and velocities. For comparison, we simulate a 'No skeleton' Baymax without the support of rigid bones. Its bone error is non-zero because of the deformation. The Baymax in a differentiable rigid body simulator [62] is rigid, so the body length error is non-zero. Li et al. [45] simulate the 'Passive' skeleton case where there is no joint actuation and joint angles cannot be adjusted to the desired configuration. We also run Difftaichi-MPM [30] by converting the mesh model to the point-based MPM representation. 'MPM' does not have skeletons so the errors are high. The arms also detach from the body so the joint error is NaN. Our method attains the highest degree of physical realism and correctness overall.

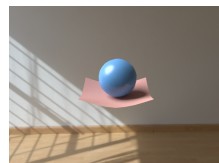

| | Penetration error | Compression | Stretching | Runtime (ms) |
|---|---|---|---|---|
| Cloth [54] | $0.041 \pm 0.004$ | No | No | $930 \pm 52$ |
| Rigid [62] | **0** | No | No | $520 \pm 135$ |
| MPM [30] | NaN | **Yes** | **Yes** | $\mathbf{14 \pm 0}$ |
| Ours | **0** | **Yes** | **Yes** | $51 \pm 3$ |

Figure 2: Ablation study of collision handling scheme. 'Penetration error' is greater than 0 if objects interpenetrate as a result of the simulation. 'Compression' and 'stretching' indicate whether the simulation allows the ball to compress and stretch in the vertical and horizontal directions, respectively. Our method can model a soft ball correctly while preventing interpenetration.

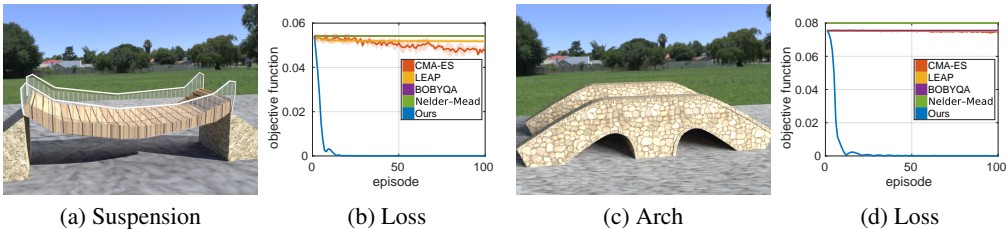

| (a) Suspension | (b) Loss | (c) Arch | (d) Loss |
|---|---|---|---|

Figure 3: System identification. Given the desired displacement under gravity, we estimate Poisson's ratio and Young's modulus of each element of the two bridges with different structures. Our method converges much faster than gradient-free methods.

**Contact handling.** Good contact handling is critical for simulating multi-body systems that interact with their environment. In this experiment, we throw a 3D soft ball against a 2D thin sheet. Metrics in this experiment are penetration error and indicators of vertical compression and horizontal stretching. Zero penetration error is ideal. 'Yes' for compression/stretching indicates that the simulator can model the deformation of the soft ball correctly. The metrics are averaged over 5 experiments with different initial positions and velocities. The dry frictional contact model of Ly et al. [54] does not model the deformation of soft solids, and there could be penetration when the resolutions of the ball and cloth differ a lot due to the nodal collision handling scheme. The rigid differentiable simulator of Qiao et al. [62] can prevent interpenetration, but the ball remains rigid. MPM [30] can model the deformation of both the ball and the cloth, but the cloth is torn apart by the ball and penetration cannot be quantified. In contrast, our method accurately handles collision to avoid interpenetration and correctly simulates the deformation of the ball.

## 6.3 Applications

**System identification.** Determining the material parameters of deformable objects can be challenging given their high dimensionality and complex dynamics. In this experiment, we use our differentiable simulator to identify the material property of each finite element cell within the soft body. As shown in Figure 3, there are two bridges with unknown materials: a suspension bridge with both ends fixed and the entire bridge being soft, and an arch bridge that has three piers attached to the ground. Given that the movement of the barycenter under gravity, compared to its rest pose, is $\Delta x = 8cm$, we estimate Young's modulus and Poisson's ratio of each finite element cell in the bridge. The loss function is the distance from the actual barycenter to the target. The suspension bridge has $n = 668$ cells and the arch one has $n = 2911$ cells. The number of unknowns is $2n$. We compare our method with four derivative-free methods (CMA-ES [25], LEAP [8], BOBYQA [61], and Nelder-Mead [66]). Each experiment is repeated 5 times with different random seeds. As shown in the figure, our method converges in $\sim 10$ iterations while others fail to converge even after 100 iterations, indicating that derivative-free methods in this high-dimensional setting become too inefficient to converge to a reasonable solution. By making use of the gradients provided by our method, common gradient-based algorithms can quickly reach the target configuration.

**Motion planning.** Controlling the motion of deformable bodies is challenging due to their flexible shapes. In this experiment, summarized in Figure 4, the task is to control robots with different actuator types. In general, given an initial state $X_0$, control policy $\phi_{\theta_1}(\cdot)$, and material parameters $\theta_2$, the simulator can generate a trajectory of the states at all time steps $t$, $\{sim_t(X_0, \phi_{\theta_1}(\cdot), \theta_2)\} = \{X_1, X_2, ..., X_n\}$. If we want the system to reach a target state $X_{target}$ at the end of the simulation, we can define an objective function $L(X_0, \theta_1, \theta_2) = ||(sim_N(X_0, \phi_{\theta_1}(\cdot), \theta_2) - X_{target})||_2$, where the optimization variable are $\theta_1$ and/or $\theta_2$.

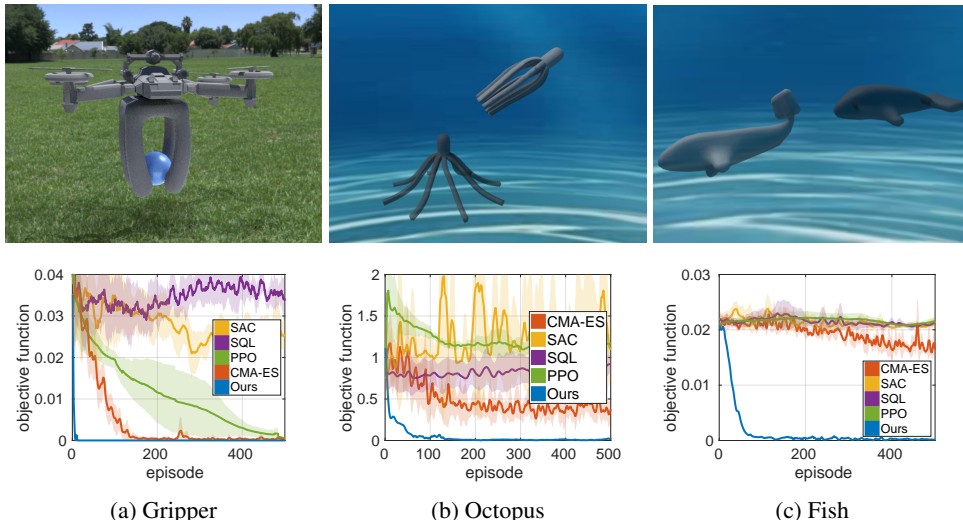

Figure 4: Motion control experiments. We simulate (a) a gripper with pneumatic cells in four arms, (b) an octopus with 16 muscles, and (c) a fish with 6 bones inside the body. The goal is to optimize the actuation schedules to get the robots to the target location. Our method is up to *orders of magnitude* faster than reinforcement learning and derivative-free baselines.

We compare our method with Reinforcement Learning algorithms (SAC [23], SQL [22], and PPO [64]), and the best derivative-free optimization method from the last experiment, CMA-ES. We also tried MBPO [35], but we found that this method takes too much memory and could not finish any test. All RL methods use the negative of the loss as the reward.

The **pneumatic gripper** in Figure 4(a) has 56 pneumatic cells in four arms and is attached to an (invisible) drone as in [17]. The pneumatic activation can control the volume of a tetrahedron. When the cells inflate, the arms will move inwards and hold the ball tighter. We control the pneumatic activation as well as the movement of the drone to move the ball from the start $(0, 0, 0)$ to our target $(0, 0.3, 0)$ in 50 steps. The loss is the distance from the actual position to the target position. Our method converges in 10 episodes while CMA-ES and PPO gradually converge in 200 and 500 episodes, respectively.

The **muscle-driven octopus** in Figure 4(b) has 8 legs, each with 2 muscles inside. It moves forward by actuating the muscles, being pushed by drag and thrust forces induced by the water on the octopus's surface [59]. The octopus starts at $(0, 0, 0)$ and our target location is $(-0.4, 0.8, -0.4)$. We set the objective to be the distance between the current location and the desired location. The length of the simulation is 400 steps, and the control input in each step is 64-dimensional. In total, there are $64 \times 400 = 25600$ variables to optimize. Our method converges in 50 episodes while other methods fail to converge in 500 episodes.

The fish with an **embedded skeleton** in Figure 4(c) has 6 bones: 3 in its body, 2 in the fins, and 1 in the tail. The hydrodynamics in this environment is the same as in the octopus experiment. The fish starts at $(0, 0, 0)$ in step 1 and the target location in step 100 is $(0, 0, 0.15)$. The objective function is the distance from the actual location to the target location. For each step, there will be a torque vector of size 5 that represents the joint actuation level. In total, the optimization variable has 500 dimensions. Our method with gradient-based optimization can converge in roughly 50 episodes, while others cannot converge even after 500 episodes.

In summary, gradient-free optimization methods and RL algorithms meet substantial difficulties when tackling problems with high dimensionality, such as soft, multi-body systems. Even when the action space is as small as the one in the gripper case, RL methods still fail to rapidly optimize the policy. By introducing the gradients of the simulation, simple gradient-based optimization outperforms other algorithms. This work hopefully may inspire improvements in RL algorithms that tackle such high-dimensional problems.

## 7 Conclusion

Our paper has developed a differentiable physics framework for soft, articulated bodies with dry frictional contact. To make the simulation realistic and easy-to-use, we designed a recursive matrix

assembly algorithm and a generalized dry frictional model for soft continuum with a new matrix splitting strategy. Integrated with joint, muscle, and pneumatic actuators, our method can simulate a variety of soft robots. Using our differentiable physics to enable gradient-based optimization, our method converges more than an order of magnitude faster than the baselines and other existing alternatives.

There are some limitations in our contact handling and soft body dynamics. Currently, though our algorithm is more extensive and generalized than existing differentiable physics algorithms and our implementation handles the most commonly found contact configuration, vertex-face collisions, there could still be edge-edge penetration missed in some corner cases. Moreover, the Projective Dynamics pipeline limits the energy to have the form $E = \|\mathbf{Gq} - \mathbf{p}\|$. Some nonlinear material models (e.g., neo-Hookean) are not captured in this framework and new models for differentiable physics will be required for handling nonlinear and heterogeneous materials. For future work, we aim to add edge-edge collision handling in the Projective Dynamics pipeline. The techniques in [53] can be used to incorporate addition material types. GPU or other parallel computing implementation can be used to boost the performance of gradient computation.

**Acknowledgements.** This research is supported in part by Army Research Office, National Science Foundation, Dr. Barry Mersky and Capital One Endowed Professorship.

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
