# Supplementary Information

## Appendix A  Proof for Proposition 1

Below we prove Proposition 1. For ease of analysis, we assume the soft body has only one tetrahedron.

According to our assumption, we have:

$$\frac{\mathbf{M}^x}{h^2} = \begin{bmatrix} \frac{\rho V}{4h^2} & & & \\ & \frac{\rho V}{4h^2} & & \\ & & \frac{\rho V}{4h^2} & \\ & & & \frac{\rho V}{4h^2} \end{bmatrix} \qquad \mathbf{L}^x = \mu V \mathbf{A}^{x\top} \mathbf{A}^x \tag{22}$$

$$\mathbf{A}^x = \begin{bmatrix} -\mathbf{P}^{-\top}\mathbf{s} & \mathbf{P}^{-\top} \end{bmatrix} \qquad \mathbf{P} = \begin{bmatrix} \mathbf{q}_1 - \mathbf{q}_0 & \mathbf{q}_2 - \mathbf{q}_0 & \mathbf{q}_3 - \mathbf{q}_0 \end{bmatrix} \qquad \mathbf{s} = \begin{pmatrix} 1 & 1 & 1 \end{pmatrix}^\top \tag{23}$$

where $*_x$ means the submatrix of the original one after extracting the rows and columns that represents the x axis (we omit the other axes because the coefficients are the same), $\rho$ is the density, $V$ the tetrahedron volume, $\mu$ the material stiffness, $\mathbf{P}$ the displacement matrix for the tetrahedron, and $\mathbf{p}_i$ the coordinates of the $i^{\text{th}}$ vertex. Here, $\mathbf{A}_x$ can be regarded as the spatial differential operator, which is usually used to compute the deformation gradient of the soft body.

Assuming fixed forces of $\mathbf{k}$ and $\xi$ in Eq. 19, the iterative method becomes the Jacobi method, whose convergence is determined by the spectral radius $\rho(\mathbf{M}^{-1}\mathbf{L})$. We relax this condition to obtain a sufficient condition: if the diagonals of $\mathbf{M}$ is larger than the row sums of $\mathbf{L}$, the Jacobi method is guaranteed to converge.

By definition we have

$$\frac{\rho V}{4h^2} > \sum_j |\mathbf{L}_{ij}^x| \tag{24}$$

Since $\mathbf{L}_{ij} = \mu V \langle \mathbf{A}_{*i}, \mathbf{A}_{*j} \rangle$, we have

$$\frac{\rho}{4h^2\mu} > \sum_j |\langle \mathbf{a}_i, \mathbf{a}_j \rangle| \geq 6\|\mathbf{P}^{-1}\mathbf{P}^{-\top}\|_\infty \tag{25}$$

since $\mathbf{A}^{x\top}\mathbf{A}^x = \begin{bmatrix} \mathbf{s}^\top\mathbf{P}^{-1}\mathbf{P}^{-\top}\mathbf{s} & -\mathbf{s}\mathbf{P}^{-1}\mathbf{P}^{-\top} \\ -\mathbf{P}^{-1}\mathbf{P}^{-\top}\mathbf{s} & \mathbf{P}^{-1}\mathbf{P}^{-\top} \end{bmatrix}$. Further reducing the right hand side yields:

$$\frac{\rho}{24h^2\mu} > \|\mathbf{P}^{-1}\mathbf{P}^{-\top}\|_\infty \geq \frac{1}{\sqrt{3}}\|\mathbf{P}^{-1}\mathbf{P}^{-\top}\|_2 = \frac{1}{\sqrt{3}}\lambda_{min}^{-1}(\mathbf{P}^\top\mathbf{P}) \tag{26}$$

where $\lambda_{min}$ denotes the minimum eigenvalue of the matrix. It is then straightforward to obtain the inequality with respect to the edge lengths of the tetrahedron:

$$\lambda_{min}(\mathbf{P}^\top\mathbf{P}) \leq \frac{1}{3}\text{tr}\,\mathbf{P}^\top\mathbf{P} = \frac{1}{3}\sum_{k=1}^3 \|\mathbf{q}_k - \mathbf{q}_0\|_2^2 \tag{27}$$

To sum up, we reach a condition that constrains the upper bound of $h$:

$$h^2 < \frac{\rho}{24\sqrt{3}\mu\sum_{k=1}^3 \|\mathbf{q}_k - \mathbf{q}_0\|_2^2} \tag{28}$$

In fact, it is easy to extend the results to a soft body system with multiple tetrahedrons. All we need to to is to sum up the $\mathbf{L}$ matrix for all individual elements in Eq. 24 while increasing the mass matrix by a small constant factor (from $\rho/4$ to $O(1)\rho$, so we omit it for convenience). The upper bound of $h$ of the original method becomes even smaller:

$$h^2 < \frac{\rho}{24\sqrt{3}T\mu\sum_{k=1}^3 \|\mathbf{q}_k - \mathbf{q}_0\|_2^2} \tag{29}$$

where $T$ is the number of tetrahedrons. We assume the soft body has $T$ identical tetrahedrons for analysis convenience. For general cases, one can simply replace $T\sum_{k=1}^3 \|\mathbf{q}_k - \mathbf{q}_0\|_2^2$ with $\sum_i \sum_{k=1}^3 \|\mathbf{q}_{i,k} - \mathbf{q}_{i,0}\|_2^2$ where $i$ is the index of tetrahedrons. Using the setting in our experiments, where $T \approx 1000$, $\mu \approx 3 \times 10^5$, $\|\mathbf{q}_k - \mathbf{q}_0\|_2 \approx 10^{-2}$, and $\rho \approx 1$, we would need to set $h < 1/1934$ in order to ensure convergence.

## Appendix B  Convergence Proof for Eq. 21 in Splitting Scheme

Below we prove that Eq. 21 is guaranteed to converge. Having the same assumption as Appendix A, the convergence now is determined by the spectral radius $\rho((\mathbf{M} + \mathbf{D})^{-1}(\mathbf{L} - \mathbf{D}))$. We relax this condition to

obtain a sufficient condition: if $\mathbf{M} + \mathbf{L}$ is strictly row diagonally dominating, the Jacobi method is guaranteed to converge.:

$$\frac{\rho V}{4h^2} + |\mathbf{L}_{ii}^x| > \sum_{i \neq j} |\mathbf{L}_{ij}^x| \tag{30}$$

for all $1 \leq i \leq 4$, yielded by the definition of row diagonally dominance. Since $\mathbf{L}_{ij} = \mu V \langle \mathbf{A}_{*i}, \mathbf{A}_{*j} \rangle$, we have

$$\frac{\rho}{4h^2\mu} > \sum_{i \neq j} |\langle \mathbf{a}_i, \mathbf{a}_j \rangle| - |\langle \mathbf{a}_i, \mathbf{a}_i \rangle| \geq 0 \tag{31}$$

given that $\sum_j \langle \mathbf{a}_i, \mathbf{a}_j \rangle = 0$ since the row sum of $\mathbf{A}^x$ is $\mathbf{0}$. This completes the proof.

To extend this conclusion to multiple tetrahedrons, we use the same method discussed before: we sum up the $\mathbf{L}$ matrix for all individual elements in Eq. 30. Since the right hand side is 0, summing up the inequalities will not change the conclusion; the convergence is still guaranteed.

## Appendix C    Joints

We give two examples here: 1-DoF rotational joint and prismatic joint.

**Rotational joint.** This joint is characterized by a rotation axis $\mathbf{n}$ and the angle $\theta$. Its transformation matrix and the Jacobian are:

$$\mathbf{A}^r = \begin{bmatrix} \mathbf{R} & \mathbf{0} \\ \mathbf{0} & 1 \end{bmatrix} \qquad \frac{\partial \mathbf{A}^r}{\partial \theta} = \begin{bmatrix} \frac{\partial \mathbf{R}}{\partial \theta} & \mathbf{0} \\ \mathbf{0} & 0 \end{bmatrix} \tag{32}$$

$$\mathbf{R} = \cos\theta \cdot \mathbf{I} + \sin\theta [\mathbf{n}]_\times + (1 - \cos\theta)\mathbf{n}\mathbf{n}^\top \tag{33}$$

$$\frac{\partial \mathbf{R}}{\partial \theta} = -\sin\theta \cdot \mathbf{I} + \cos\theta [\mathbf{n}]_\times + \sin\theta \mathbf{n}\mathbf{n}^\top \tag{34}$$

The local update of the rotational joint is given by:

$$\theta^{i+1} = \arctan(\sin\theta^i + \cos\theta^i \Delta\theta^i, \cos\theta^i - \sin\theta^i \Delta\theta^i) \tag{35}$$

**Prismatic joint.** This joint is characterized by a prismatic axis $\mathbf{u}$ and the scale $l$. Its transformation matrix and the Jacobian are:

$$\mathbf{A}^p = \begin{bmatrix} \mathbf{I} & l\mathbf{u} \\ \mathbf{0} & 1 \end{bmatrix} \qquad \frac{\partial \mathbf{A}^p}{\partial l} = \begin{bmatrix} \mathbf{0} & \mathbf{u} \\ \mathbf{0} & 0 \end{bmatrix} \tag{36}$$

$$\tag{37}$$

The local update of the prismatic joint is simply addition:

$$l^{i+1} = l^i + \Delta l^i \tag{38}$$

## Appendix D    Actuators

**Pneumatic actuator.** We use co-rotational elastic strain energy model for tetrahedral cells. For a pneumatic cell with activation level $a$, the energy is computed as

$$\Psi_{pneumatic}(\mathbf{F}, a) = \frac{k_p}{2} \|\mathbf{F} - \mathbf{R}(a)\|^2 \tag{39}$$

To compute $\mathbf{R}(a)$, we first perform SVD decomposition on the deformation gradient $\mathbf{F} = \mathbf{U}\Sigma\mathbf{V}^T$. Then $\mathbf{R}(a)$ can be written as $\mathbf{R}(a) = \mathbf{U}\Sigma^*\mathbf{V}^T$, where $\Sigma^* = \mathbf{D} + \Sigma$, and $\mathbf{D}$ is computed by

$$\arg\min_{\mathbf{D}} \|\mathbf{D}\|_2^2, s.t. \prod_i (\Sigma_i i + \mathbf{D}_i) = a \tag{40}$$

**Muscle actuator.** We use the muscle actuators described in [59]. Muscles are modeled as fibers in the soft bodies, and the forces are computed as $\mathbf{f}_{muscle}(a) = -f_{muscle}(a)\mathbf{m}$, where $a \in [0, 1]$ is the activation level, $\mathbf{m}$ is the direction of fiber. To implement this force, a strain energy model [41] is used, $E_{muscle} = \mathbf{V}_{muscle}\Psi_{muscle}(\mathbf{F}, e)$, where $\Psi_{muscle}(\mathbf{F}, a) = \frac{k_m}{2}\|(1 - r)\mathbf{F}\mathbf{m}\|$, $k_m$ is the stiffness, $r = \frac{1-a}{l}$ is the projection of the cord segment, $l = \|\mathbf{F}\mathbf{m}\|$ is the stretch factor.

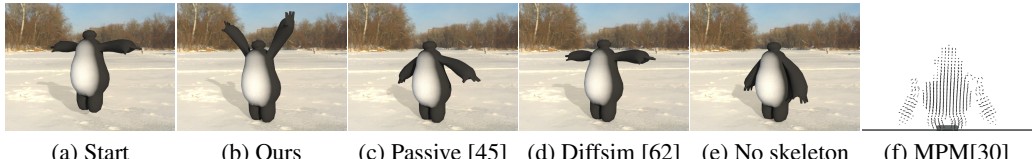

| (a) Start | (b) Ours | (c) Passive [45] | (d) Diffsim [62] | (e) No skeleton | (f) MPM[30] |

Figure 5: Ablation study of embedded skeletons. (a) is the initial frame where a Baymax is released from the air and will hit the ground. (b) is a baymax with bones, torques are applied to two shoulder joints so it lifts the arms. (c) is simulated by passive skeleton [45]. (d) Baymax in the differentiable rigid body simulator [62] does not have deformation. Baymax has no skeletons in (e), where the arms are stretched and the legs are collapsed. The MPM particles scatters after the collision in (f).

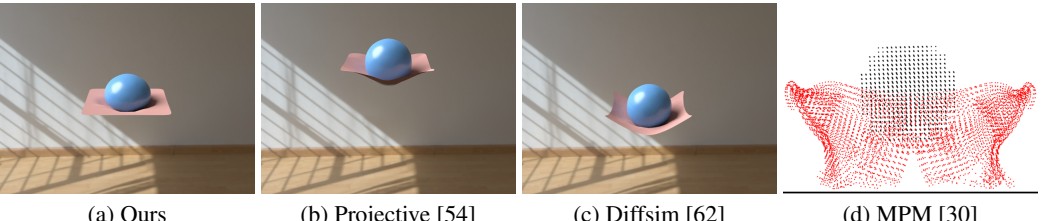

| (a) Ours | (b) Projective [54] | (c) Diffsim [62] | (d) MPM [30] |

Figure 6: Ablation study of collision handling scheme. A ball collides with a cloth. Our method in (a) can simulate the deformation of both objects. The dry frictional model for cloth simulates (b), where the solid ball is modeled as rigid body and there is penetration because of the nodal collision handling. The ball is also modeled as a rigid body by [62] in (c). The cloth is teared apart when modeled as MPM particles in (d).

## Appendix E   Ablation study

Figure 5 shows the ablation study of skeletons. In the experiment, a Baymax in its T-pose is released above the ground, as shown in Figure 5(a). In (b), we embed 5 bones inside the Baymax (4 in arms and legs, and 1 in the torso). When the Baymax falls to the ground, we also add torques on its shoulders so it can lift its arms. (c) shows the result of [45], where the skeleton is passive and we cannot apply torques. (d) is simulated by another differentiable simulator, Diffsim [62]. It can only simulate cloth and rigid bodies, so Baymax maintains the rest pose. (e) is simulated without a skeleton. Without the support of rigid bones, we notice that its arms are stretched and its legs are compressed. We also run Difftaichi-MPM in (d), where the arms of the Baymax are detached from the body after collision.

Figure 6 is the ablation study for different contact handling schemes. A soft ball is released from the air and then collide with a cloth. (a) is our simulation results, where both the ball and the cloth have signification deformation. We run the dry frictional contact algorithm designed for cloth [54] in (b), but the method cannot simulate 3D soft objects, so the ball is rigid. And there is also slight penetration due to the instability of nodal collision handling. (c) is Diffsim [62], which can simulate cloth and rigid body, but the ball is not differentiable. (d) is simulated by Difftaichi-MPM. The objects can deform, but the elasticity is not realistic and the cloth will break down after the collision.