# OpenReview forum: "Differentiable Simulation of Soft Multi-body Systems"
_NeurIPS.cc/2021/Conference — NeurIPS 2021 Poster_

### Official Review · Reviewer_JAVC · 2021-06-28

**Rating:** 7
**Confidence:** 4

**Summary:**

The proposed paper introduces a novel method for solving the simulation of soft articulated bodies based on a differentiable solver. The key contribution of the introduced method are a top-down matrix assembly approach for projective dynamics, a generalized friction model, analytic models for muscles, and a unified framework that enables computing gradients in a differentiable manner. The method is evaluated based on a number of meaningful experiments and ablation studies.

**Ethical Concerns:**

I do not have any ethical or diversity concerns toward the topic this paper addresses or the paper in general.

**Limitations And Societal Impact:**

Limitations of the method are appropriately discussed. I do not have concerns that the method may have potential negative societal impact.

**Main Review:**

Overall, this is an interesting paper that has all things going for it. The method is technically sound, the paper is well-written, and the experiment setup along with the presented results are convincing. As also stated in the paper, the research direction the paper addresses is important for robotics and research in the direction of differentiable physics is potentially of high impact. W.r.t. the paper I specifically like the experiments towards system identification and motion planning. So, in summary, I am in support of accepting this work to NeurIPS.


Specific comments:

- At the beginning of Section 3 it would be helpful to first formally define the geometric representation (tetrahedral meshes). Also, at the beginning of Section 3 it is not clear why vertices are defined per frame. Some introducing/clarifying sentences would be required here.

- It would also be helpful to discuss what implications exists due to relying on tetrahedral meshes. Can the method also be used for other (less constrained) geometric representations?

- Section 3: I would also suggest to move the introduction to Projective Dynamics to the Appendix. Equations (1)-(3) are either direct copies or reformulation and given that reference [5] is already provided, restating them here does not seem necessary. Similar for Algorithm 1. For Equation (4)-(5) it would be helpful to more clearly describe what the differences to [5] are.

- Section 4: Here it would also help to first motivate why it is necessary to add articulated skeletons prior to stating that it is challenging to add them.

- L148,L149: It is not clear what is meant by ' ... if the rigid body vertices are ordered
149 by their kinematic tree depth.'.

- L157: Please clearly state what 'the tree' is. Currently it is not defined.

- L180: Missing ' ' between 'friction.The idea'

- I appreciated the discussion in L306-311. It would even be interesting to further expand on the reasons as to why the algorithms perform as observed.

- Given the results shown, the statement that the 'method is able to simulate a variety of soft robots' seems somewhat unsubstantiated and should be toned-down.

The authors may want to add the following papers to their discussion of related work on deformable body simulation and differentiable physics:

- T. Hädrich, B. Benes, O. Deussen, S. Pirk, Interactive Modeling and Authoring of Climbing Plants, Computer Graphics Forum (Proceedings of Eurographics), 2017

- H. Shao, T. Kugelstadt, W. Palubicki, J. Bender, S. Pirk, D. L. Michels, Accurately Solving Physical Systems with Graph Learning, 2020

**Time Spent Reviewing:**

3 hours.

---

> ### Author Response · Authors · 2021-08-09
> **Response to Reviewer JAVC**
>
> Q1: At the beginning of Section 3 it would be helpful to first formally define the geometric representation (tetrahedral meshes). Also, at the beginning of Section 3, it is not clear why vertices are defined per frame. Some introducing/clarifying sentences would be required here.
>
> A1: Tetrahedron meshes are one kind of discretization of 3D objects where the basic elements are tetrahedrons. Such data structure is described by its vertices and triangle faces. Since the soft body system is moving, the vertex locations are changing across frames, thus the variables to record the locations in each frame. We are happy to add the details as suggested.
>
>
> Q2: What implications exist due to relying on tetrahedral meshes. Can the method also be used for other (less constrained) geometric representations?
>
> A2: The form of the energy term depends on the geometry representation. If we change the tetrahedral meshes to other representations, like volume mesh, the form of $G_i$ and $p_i$ in Equation (3) will also change. But the entire pipeline still works. For more specialized dynamics like a rod, yarn, and fluids, though tetrahedral meshes would still be applicable, it may be preferable to use other specialized
> representations for higher efficiency based on the chosen application domains.
>
> Q3: It might be better to move the introduction to Projective Dynamics to the Appendix.
>
> A3:  We considered doing the same, but finally decided to keep it there to make the paper self-contained. Moreover, to describe our novel techniques in Sections 4 and 5, we use notations from defined in Section 3 on Projective Dynamics, without these notations the paper would be
> difficult to understand for general readers not knowing Projective Dynamics before.
>
> Q4: Why is it necessary to add articulated skeletons?
>
> A4: Thank you for asking. Besides the discussions about skeletons in Sections 1 and 2. We will add our motivation to Section 4, "Skeletons are indispensable for vertebrate animals and nowadays articulated robots.  However, adding skeletons into the soft body simulation is challenging."
>
> Q5: Please clearly state what 'the tree' is.
>
> A5: We will add a figure to explain the kinematic tree. For example, we can view the skeleton of a Baymax as a tree. The Baymax's head is the root link, and its child link is the torso. The four limbs are leaf links, whose parent link is the torso.  An added figure will illustrate this point
> more clearly.
>
> Q6: It is not clear what is meant by `... if the rigid body vertices are ordered by their kinematic tree depth.'.
>
> A6: In this sentence in Line 149, we want to provide a general idea how the matrix B looks like. B could be reordered into a block lower triangular matrix. If we sort the vertices by their depth as in the kinematic tree, the position of one vertex $i$ at the link $k$ only depends on its ancestor links. Therefore, in the matrix B, the entries corresponding to $(i,1),...,(i,k)$ are non-zero while entries $(i,k+1),...,(i,n)$ are zero (assume that the link 1 is the root and the link $n$ is the leaf).  Perhaps a figure would be helpful here as well.
>
> Q7: In L306, why do the algorithms perform as observed?
>
> A7: Our hypothesis is that even though the action space is small, the DoF of the object is too high. We cannot directly use all DoFs of a soft body system as the observation state due to memory limits. Instead, we can only use some partial observation (e.g. keypoints and average speed) as input signals, which limit the capacity of the RL algorithm.
>
> Q8: Related works.
>
> A8: Thank you for the suggestion.  We will add the mentioned related papers. [1] models the dynamics of climbing plants and allows interactive editing. [2] uses graph network to learn the initial guess of rod physics solvers thus accelerating convergence.
>
> [1] T. Hädrich, B. Benes, O. Deussen, S. Pirk, Interactive Modeling and Authoring of Climbing Plants, Computer GraphicsForum (Proceedings of Eurographics), 2017
>
> [2] H. Shao, T. Kugelstadt, W. Palubicki, J. Bender, S. Pirk, D. L. Michels, Accurately Solving Physical Systems with GraphLearning, 2020

---

> > ### Comment · Reviewer_JAVC · 2021-09-03
> > **Thank you**
> >
> > Thank you for your responses. After reading the other reviews and the corresponding rebuttal statements I still think this is an interesting paper and I support acceptance. Please carefully address the comments for the final version of the manuscript.

---

> > > ### Author Response · Authors · 2021-09-03
> > > **Comments will be addressed for the revised version of the manuscript.**
> > >
> > > Thank you so much for all your suggestions and questions. We will address the comments and update the changes in our rebuttal to the revised version.

---

### Official Review · Reviewer_yjag · 2021-07-10

**Rating:** 7
**Confidence:** 3

**Summary:**

The paper presents a new soft-body physics simulator that is end-to-end differentiable, and shows how it supports efficient system identification and control.  Experiments show the value of the simulator for a range of complex realistic situations -- although only in simulation.  The experiments also report ablations of key model components (skeletons, collision contacts).

**Limitations And Societal Impact:**

Limitations are described briefly at the end.  That's okay but I would encourage the authors to illustrate concretely (perhaps in the supplementary material) some failure cases and limitations of the current method, which point the way to future work.  I would be especially interested to see how the system breaks down when the contact model isn't completely right.

**Main Review:**

I am not an expert in the technical aspect of differentiable simulation or soft-body physics, so my comments should be taken with an appropriate grain of salt.  But I have collaborated as a senior co-author on several projects using related simulators for AI and machine learning problem.  I liked this paper a lot: The topic is important, the writing is clear, and the contribution meaningful.  The experiments are well-done and well-described.  I can't judge originality like an expert in simulation could, but my impression is that the method is novel enough for a top conference contribution. I expect this work will be of interest to a number of different sub-communities at NeurIPS: RL, robotics, and maybe computer vision researchers interested in intuitive physics, and possibly some neuroscientists as well.

The main thing I felt was missing is an evaluation of the system on scenarios that don't match the physics of the simulator: this could be either control of a simple real-world robot, or control or system identification in a simulator that has different physics than the differentiable simulator (i.e., simulating deployment of the model-based methods in the paper to real-world systems where there is some model mismatch).  I don't see this as a fatal flaw but it would have been very nice to see.  Anything like this that could be added in revision might lead me to raise my score.

**Time Spent Reviewing:**

1

---

> ### Author Response · Authors · 2021-08-09
> **Response to Reviewer yjag**
>
> Q1: Simulating deployment of the model-based methods in the paper to real-world systems where there is some model mismatch.
>
> A1: Good point. We hope our method can be used to analyze robots or real creatures. But deploying algorithms on real robots requires additional hardware integration and engineering efforts, which is beyond the scope of this paper.   But, this is definitely an exciting future work that we would like to explore next.
>
> Q2: Illustrate concretely (perhaps in the supplementary material) some failure cases and limitations of the current method, which point the way to future work.
>
> A2: The edge-edge collision can lead to failure cases, as mentioned in Line 326. If the resolution of the simulated mesh is low, the penetration can be visible. Imagine two big tetrahedrons collide at their edges. Even though their vertices will not penetrate, a segment of their edges can go into each other's interior.
> Another promising future work is to make the collision resolution more robust to handle challenging cases such as multi-layer dense contacts so as to enable a wider range of applications.
> We will illustrate such examples in the supplementary materials, as suggested.

---

### Official Review · Reviewer_PGkZ · 2021-07-16

**Rating:** 6
**Confidence:** 2

**Summary:**

This paper proposes a soft multi-body differentiable physics simulation framework based on projective dynamics. The authors propose a top-down matrix assembly algorithm for rigid body simulation algorithms and a new matrix splitting strategy for a generalized dry friction model for soft continuum. The experiments demonstrate that their designs make soft body simulation more stable and realistic compared to other frameworks, and the gradients help accelerate system identification and motion control of soft robots.

**Limitations And Societal Impact:**

Yes

**Main Review:**

Differentiation of projective dynamics and simulation of an articulated soft body system are all noticeable features of the proposed framework and would benefit the machine learning and robotics community. The authors also provide code supplementarily. I would be happy to see it at the Nuerips conference. I am only a little bit worry if this paper fits Neurips's general audience. The paper focuses more on the simulation side, which might be better evaluated in other relevant conferences such as Siggraph or other soft robot conferences. Here are some comments:

- Algorithm 2 computes the Jacobian of each link. I wonder what is the differences between the proposed algorithm and the way of computing jacobian in rigid body simulation?
- Section 5 mentions a novel matrix splitting method and shows that in theory it convergences much faster. I wonder how this will affect the experimental/visual effects. I hope the authors could include experiments to justify their contributions.
- What's the efficiency of the proposed algorithm? How is it compared with MPM implemented with Taichi? Is it fast enough for RL research?

**Time Spent Reviewing:**

4

---

> ### Author Response · Authors · 2021-08-09
> **Response to Reviewer PGkZ**
>
> Q1: What are the differences between the proposed algorithm and the way of computing Jacobian in rigid body simulation?
>
> A1: In a rigid body, the Jacobian is only related to the transformation and rotation of the rigid body (6 degrees of freedom). For articulated bodies, however, we need to calculate the Jacobian w.r.t. all the parent links' degree of freedom, which is computed iteratively.
>
> Q2: Section 5 mentions a novel matrix splitting method and shows that in theory it convergences much faster. I wonderhow this will affect the experimental/visual effects.
>
> A2: The splitting strategy allows us to use larger time steps while still guarantee convergence. For example, if we want to simulate a 1-second motion, our method can run 200 frames with the time step $h=5\times 10^{-3}s$ while the solver without the splitting strategy needs to run 2000 frames with the time step $h=5\times 10^{-4}s$. Since the time for running one frame is fixed, this splitting strategy can achieve 10x shorter running time in this case, thus making the resulting animation appear more natural and appealing in \textit{real time} without lags due to long computation time.
>
> Q3: What's the efficiency of the proposed algorithm? How is it compared with MPM implemented with Taichi? Is it fast enough for RL research?
>
> A3: We report the runtime in Figure 2.  While Taichi is 14 ms per frame and our method is 51 ms per frame, our approach is fast enough for real-time simulation ($\sim$20 fps) and RL research. Taichi is slightly faster since their dynamics of MPM is different from our more complex dynamical system.  Also, it is implemented on GPU, while ours runs on CPU.  (GPU implementation can provide roughly an order of magnitude runtime performance boost.)

---

### Official Review · Reviewer_yZqv · 2021-07-17

**Rating:** 7
**Confidence:** 3

**Summary:**

This paper takes on the timely problem of devising differentiable simulators for soft bodies, with a focus on articulated bodies where the soft nature of the material comes with structure in the way different parts are connected together - hence requiring formulation of equations of motion in the sense of multi-body dynamics.

The core contribution is to formulate the simulation problem in a projective dynamics framework, and to exploit the implicit matrix structure in the equations of motion to achieve computational speedups.

This has been implemented in C++ libraries and demonstrated in a few different simulation settings, including a Baymax soft articulated body, a deformable ball falling on cloth and a bridge -  each representing a different form of structure. The main outcome is that this form of differentiable simulation enables faster system identification and control.

**Ethical Concerns:**

This paper is mainly a technical contribution to a type of solver. So, there are no noteworthy ethical implications, and the authors have diligently completed the relevant sections - including code disclosure.

**Limitations And Societal Impact:**

This is mainly a technical contribution to solver methodology. It addresses specific weaknesses in earlier solvers and proposes new computational methods.

This is of course relevant to practice - indeed, the entire rationale is to enable better simulation in practical domains. However, taken by itself, this contribution does not have any negative impacts beyond the toolchains for graphics and simulation as currently used.

**Main Review:**

I think this is a timely piece of work, and it is well motivated. The authors have situated their work in context by referencing the relevant prior works - this is a large area (i.e., there are lots of papers on multi-body sim, FEM and so on), so I am content that they have referenced significant representative works to discuss different options. This includes recent interest in neural approximations, which do not really guarantee physical realism always. Also, I find the derivations to be sound.

That said, I find that the following could be better developed and explained in the paper:

(1) The experiment section is interesting in that the authors seem to have made a good attempt at comprehensively selecting different scenarios and teasing apart the reasons why their approach is better. However, some areas are lacking in detail. For instance, a crucial claim is that "this approach" outperforms RL and derivative free optimization. However, what is "this approach" in specific detail? The paper is almost entirely about the equations of motion and ways of simulating multi-body dynamics up to sec 4/5. Then, we do not have an optimization problem written down to say what the approach is for using the sim to synthesise motion. I can imagine doing this by a one-shot trajectory optimization over a horizon, I can do this by MPC, and so on - what is actually done (I suspect the former but is this stated?). In this sense, the point is not that this method outperforms RL but that a policy synthesis method with this sim as model performs better than policy synthesis without. Is this a fair characterisation of the claims (which I believe, but it isn't clearly stated)?

(2) Likewise, I find the system identification experiment interesting, because this is not just identification of a bulk modulus for the whole object but element by element Young's modulus and Poisson's ratio estimation - which is nice. It is good to see that the proposed method does a good job of quickly identifying this. However, here again I am a little bit confused - "our method" is shown to do better than derivative free estimation, but what is the framing of the optimization or error using the projected dynamics and frictions models developed in sec 4?
In this sense, the writing of sec 6 seems incomplete, and could benefit from revisiting.

(3) Without rederiving everything from first principles, I have studied the mathematical formulation - which seems sound. However, a few steps could be expanded on. For instance, after eq 7, we are told that non-rigid part of the vertices can be integrated by setting \frac{\partial{q^i}}{\partial{z}} to identity. However, non-rigid dynamics will require additional consistency conditions, e.g., rod or rope dynamics - how does that factor into all this?

(4) Lastly, it isn't entirely clear from the writing which parts of the paper are entirely original and which parts are a competent reimplementation of well understood ideas in mechanics. For instance, ideas like splitting schemes have a long tradition in applied mathematics, and the authors have drawn on this - which is great. However, it would be helpful to flag up in the prose where the novel departures are (beyond the list at the end of sec 1).


**Time Spent Reviewing:**

2

---

> ### Author Response · Authors · 2021-08-09
> **Response to Reviewer yZqv**
>
> Q1: What is the optimization problem?
>
> A1: Thanks! We will present the formulation of the optimization to make it clear.
> In general, given an initial state $X_0$, control policy $\phi_{\theta_1}(\cdot)$, and material parameters $\theta_2$, the simulator can generate a trajectory of the states at all time steps $t$, $\{sim_t(X_0, \phi_{\theta_1}(\cdot), \theta_2)\}=\{X_1, X_2, ..., X_n\}$. If we want the system to reach a target state $X_{target}$ at the end of the simulation, we can define an objective function $$L(X_0, \theta_1, \theta_2) = ||(sim_n(X_0, \phi_{\theta_1}(\cdot), \theta_2)-X_{target})||_2.$$
>
>
> Q2: What does “this approach” refer to?
>
> A2: For example, given the objective function written above, to optimize the material parameter, the problem can be written as $\arg\min_{\theta_2} L(X_0, \theta_1, \theta_2)$. Our simulator can compute the derivatives $d L / d \theta_1$, $d L / d \theta_2$, and $d L / d X_0$. When solving the problem, “this approach” refers to using gradient-based optimization methods (e.g. Gradient Descent) with the derivatives provided by our differentiable simulator to optimize the objective function. Normally, physics engines do not provide analytic derivatives, so people can only use approaches that do not rely on gradients like derivative-free optimization or reinforcement learning. Compared to those approaches, our gradient-based optimization is more efficient.
>
> Q3: Non-rigid dynamics will also require additional consistency conditions.
>
> A3: Consistency conditions (or constraints) for non-rigid dynamics are included in the energy terms in Equation (3). For more specialized applications like rod- or rope-like problems, it might be more effective to design a new simulator specifically for their dynamics, which could be interesting future work.
>
> Q4: It would be helpful to flag up in the prose where the novel departures are.
>
> A4: Thanks for the suggestion. We will add an overview for this. Section 3 is basically a preliminary of projective dynamics to explain the high-level simulation framework and define notations that will be used later. Section 4 and 5 are about how to deal with articulated skeleton and contact in our method. Splitting schemes like LU decomposition are common in numerical methods, but our scheme of splitting the large diagonal elements is specifically designed for our method.

---

> > ### Comment · Reviewer_yZqv · 2021-08-27
> > **Thanks**
> >
> > Thanks for these responses. These clarifications would be essential if the paper were to be accepted.

---

> > > ### Author Response · Authors · 2021-08-27
> > > **Clarification as stated in rebuttal will be incorporated in the final version.**
> > >
> > > Thank you so much for all your questions.  They really helped us identify places where couple more sentences can significantly improve the clarity.   Explanation provided in rebuttal will be incorporated into the final version.
> > >
> > > In addition, as stated, we will release source code along with the published paper to help further advance research in related areas like RL (using a hybrid paradigm) and differentiable programming in general.
> > >
> > > Thanks again.

---

### Decision · Program_Chairs · 2021-09-27

**Decision:**

Accept (Poster)

**Comment:**

Reviewers unanimously enjoyed this paper and thought it was a timely and impactful contribution to NeurIPS, especially in reinforcement learning and robotics. Reviewers especially highlighted the soft-body model and the motion planning experiments. There were some questions about the clarity of the exposition, and so the authors are encouraged to address the issues raised by the referees before the camera ready version of the paper is due.